# Understanding the Impact of Trampling on Rodent Bones

**Yolanda Fernández-Jalvo** [1,*], **Lucía Rueda** [1,2], **Fernando Julian Fernández** [3], **Sara García-Morato** [1,4], **María Dolores Marin-Monfort** [1,5,6], **Claudia Ines Montalvo** [7], **Rodrigo Tomassini** [6], **Michael Chazan** [8,9], **Liora K. Horwitz** [10] and **Peter Andrews** [11]

[1] Museo Nacional de Ciencias Naturales (CSIC), José Gutiérrez Abascal, 2, 28006 Madrid, Spain; lucia.rueda.dominguez@gmail.com (L.R.); sagarc16@ucm.es (S.G.-M.); dores@mncn.csic.es (M.D.M.-M.)

[2] Sciences de la Vie et de l'Environnement Université de Rennes 1, 35000 Rennes, France

[3] CONICET-Grupo de Estudios en Arqueometría, Facultad de Ingeniería, Universidad de Buenos Aires (UBA), Av. Paseo Colón 850 (CP C1063ACV), Ciudad Autónoma de Buenos Aires 1063, Argentina; fernandezf77@yahoo.com.ar

[4] Facultad de Ciencias Geológicas, Departamento de Geodinámica, Estratigrafía y Paleontología, Universidad Complutense de Madrid, Jose Antonio Novais 12, 28040 Madrid, Spain

[5] Departamento de Botánica y Geología, Universidad de Valencia, Burjassot, Valencia, 28006 Madrid, Spain

[6] Instituto Geológico del Sur (INGEOSUR), Department of Geology, Universidad Nacional del Sur (UNS)-CONICET, Avenida Alem 1253, Bahía Blanca 8000, Argentina; rodrigo.tomassini@yahoo.com.ar

[7] Facultad de Ciencias Exactas y Naturales, Universidad Nacional de La Pampa, Uruguay 151, Santa Rosa 6300, Argentina; cmontalvolp@yahoo.com.ar

[8] Department of Anthropology, University of Toronto, Toronto, ON M5S 2S2, Canada; mchazan@chass.utoronto.ca

[9] Evolutionary Studies Institute, University of the Witswatersrand, Johannesburg 2000, South Africa

[10] National Natural History Collections, Faculty of Life Sciences, The Hebrew University, Jerusalem 9190401, Israel; lix1000@gmail.com

[11] The Natural History Museum, Cromwell Road, London SW7-5BD, UK; peterandrews9@icloud.com

[*] Correspondence: yfj@mncn.csic.es

**Abstract:** Experiments based on the premise of uniformitarism are an effective tool to establish patterns of taphonomic processes acting either before, or after, burial. One process that has been extensively investigated experimentally is the impact of trampling to large mammal bones. Since trampling marks caused by sedimentary friction strongly mimic cut marks made by humans using stone tools during butchery, distinguishing the origin of such modifications is especially relevant to the study of human evolution. In contrast, damage resulting from trampling on small mammal fossil bones has received less attention, despite the fact that it may solve interesting problems relating to site formation processes. While it has been observed that the impact of compression depends on the type of substrate and dryness of the skeletal elements, the fragility of small mammal bones may imply that they will break as a response to compression. Here, we have undertaken a controlled experiment using material resistance compression equipment to simulate a preliminary experiment, previously devised by one of us, on human trampling of owl pellets. Our results demonstrate that different patterns of breakage can be distinguished under wet and dry conditions in mandibles, skulls and long bones that deform or break in a consistent way. Further, small compact bones almost always remain intact, resisting breakage under compression. The pattern obtained here was applied to a Pleistocene small mammal fossil assemblage from Wonderwerk Cave (South Africa). This collection showed unusually extensive breakage and skeletal element representation that could not be entirely explained by excavation procedures or digestion by the predator. We propose that trampling was a significant factor in small mammal bone destruction at Wonderwerk Cave, partly the product of trampling caused by the raptor that introduced the microfauna into the cave, as well as by hominins and other terrestrial animals that entered the cave and trampled pellets covering the cave floor.

**Keywords:** experimental taphonomy; bone compression; microfauna; Wonderwerk Cave

## 1. Introduction

Understanding the different processes that lead to the formation of fossiliferous assemblages can be problematic as the observed taphonomic signatures are often ambiguous. This since the distinction between damage ensuing from taphonomic agents is frequently subtle and subject to equifinality, i.e., that different taphonomic agents can lead to a common end result [1]. Experiments under controlled conditions or long-term naturalistic monitoring (e.g., [2–4]) have been highlighted as the best ways to distinguish between agents since they enable us to track the more refined signatures of modification in fossil assemblages. Such experiments allow us to gain qualitative insights as to how different processes might occur or influence how fossils accumulate during site formation.

Trampling can be defined as agents treading heavily so as to beat down or crush an object. The effect of trampling would probably not have invited special interest in taphonomic studies, but three decades ago researchers reported that trampling marks and cut marks mimicked each other (e.g., [5,6]). The main surface modifications produced by trampling on bone are striations and microstriations [7–10], shiny and polished surfaces [11,12] and even notches on oblique fracture angles [13]. In order to obtain key traits to distinguish trampling damage from cut marks, and so facilitate assessment of human involvement in a given assemblage, bones of large vertebrates have been extensively studied [14,15]. However, even after these three decades, the importance of distinguishing cut marks from trampling marks is still debated and identification continues to face difficulties even by experts [14]. Machine learning has recently been proposed as a potentially successful procedure to resolve this issue [16]. Bone breakage can be an additional modification derived from trampling and in order to distinguish it from the effects of other types of compression, researchers have characterised bone breakage (e.g., [17]). They have analysed the resistance of bones from different vertebrate species to breakage [18–20], or characterised breakage from trampling versus breakage derived from other taphonomic agents [12,21–24].

Compression testing is one of the simplest methods available that have been extensively used to characterise the mechanical properties of different materials. It has been used in ecology and taphonomy to study the response of bite force of carnivores such as hyaenas [25–27], to distinguish trampling marks obtained when bones are wet and compressed against gravels [19], to evaluate bone resistance of large mammals to sudden or slow movements when bones are wet or dry [28] and to characterise traits of compression in vertebrae of fish to distinguish them from abrasion and digestion [24,29]. Apart from digestion and skeletal element representation, breakage can be used to identify the predator that produced small mammal assemblages [11]. The effects of trampling or compression on such assemblages can thus be linked to the predator that previously modified these bones during ingestion. In such cases, broken edges may be rounded by gastric juice corrosion [11]. Thus, post-depositional processes such as trampling may be superimposed on the taphonomic pattern which characterises each predator, causing distortive effects that can be identified through experimental work.

Our experiment started with modern barn owl (*Tyto alba*) pellets, a raptor known to cause minimal modifications by digestive effects and breakage by ingestion [11]. In the case of small mammals, the fragility of their skeletal elements may give the a priori impression that they will not resist compression. This has, however, never been tested aside from observations made on two nest sites of barn owl and European eagle owl (*Bubo bubo*), and a quick trampling experiment that was described by Andrews (p.10, [11]). In the later experiment, the author stepped on owl pellets and concluded that "the pattern that emerges from these results on trampling is one of breakage of skulls, reduction in numbers of maxillae, considerable loss of teeth from the jaws leading to large numbers of isolated teeth, considerable breakage of larger postcranial elements and some degree of loss, but no loss or breakage of smaller elements (calcanei, talli))". In this study we repeated these experiments under high controlled conditions to explore the patterns of breakage caused to small mammal bones under compression. Results of the compression experiments were then compared to small mammal taphonomic analyses of fossil samples from the Earlier

Stone Age levels in Wonderwerk Cave in South Africa [30] that showed extensive breakage exceeding excavation procedures or digestion by the predator.

## 2. Material and Methods

The experiment presented here tried to reproduce the results obtained by Andrews (1990) on pellets and on isolated small mammal skeletal elements using compression under controlled conditions. Uniaxial compression experiment was conducted using a Zwick/Roell Z5.0TN machine with testXpert II software, held at the Natural Sciences Museum of Madrid (MNCN-CSIC) at the Laboratory of Environmental Analyses and Experimental Taphonomy (https://www.mncn.csic.es/en/investigacion/servicios-cientifico-tecnicos/laboratory-environmental-analyses (accessed on 27 January 2022)). This uniaxial compression machine has a test load standard of 5 kN (~500 Kg), although for the present experimental study we applied a special load cell to restrict the compression forces to 500 N (~51 Kg), this being an intermediate force between previous experimental and observational samples described by Andrews (1990). This restriction was needed to limit any extra-force exerted by mistake when programming the experiment that could completely destroy these specimens. The equipment to compress the small mammal bones used the same force in the identical direction to prevent the influence of other external parameters which could influence the results. Results of damage were obtained from the compression experiments in the form of curves through the testXpert® II software of the equipment (Figure 1).

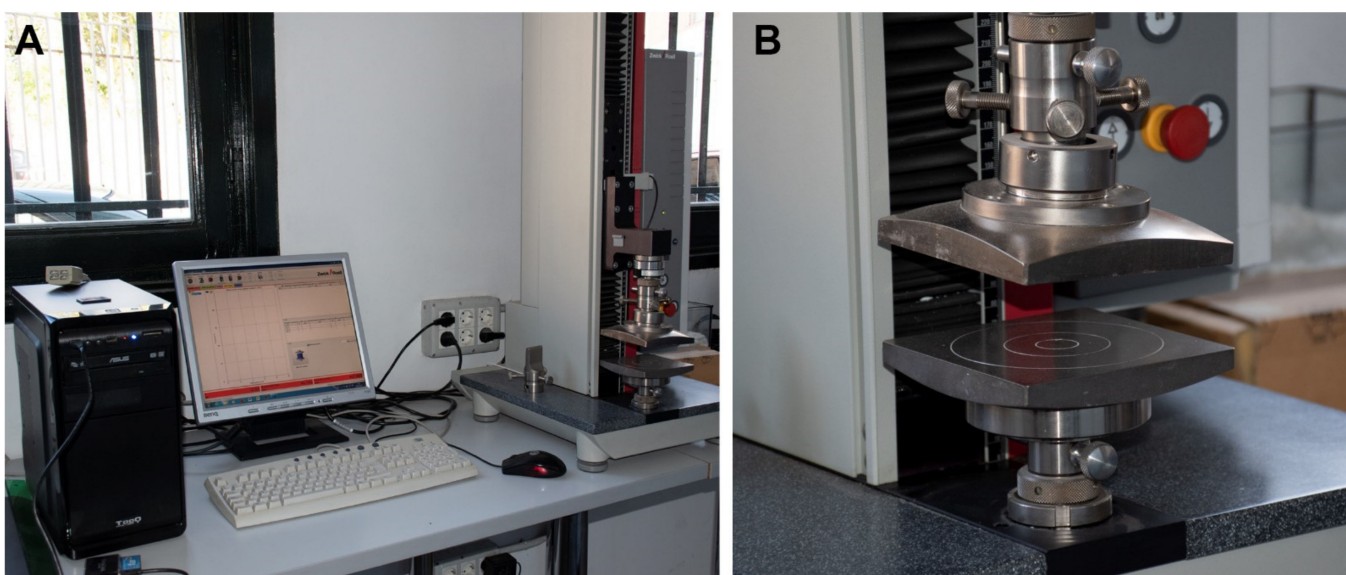

**Figure 1.** Uniaxial compression (Zwick/Roell Z5.0 TN) equipment. (**A**) Image of the equipment. (**B**) Detail of the load cell.

Small mammal materials used in the experiment derive from a modern collection of pellets of barn owls kept in captivity fed upon laboratory mice (small sized: House mouse, *Mus musculus*; large sized: Brown rat, *Rattus norvegicus*; see Table 1). Barn owl pellets are characterised by a high relative abundance of skeletal elements and low bone and tooth breakage together with a low proportion and degree of digestion on diagnostic elements (e.g., [11,31]. Despite the potential integrity of elements and abundance of skeletal remains, initial compressions performed on four pellets (to test the experiment and observations described by Andrews [11]) resulted in two of them providing only tibiae, ulnae, vertebrae and ribs, without cranial elements, while several of the skeletal elements had been affected by digestion. Eleven pellets were opened, and only five of them provided skeletal elements from cranial (jaws and skulls) and postcranial (long, square and flat bones) elements in sufficient quantity to systematically repeat the experiment several times.

**Table 1.** The 60 skeletal elements used in the experiments and their position during compression from the 8 sets considered in the experiment.

| SETS | State | Size | Skulls | Mandibles | Femora | Humeri | Pelves | Astragali | Calcanei |
|------|-------|------|--------|-----------|--------|--------|--------|-----------|----------|
| Set 1 | Dry | Large | Dorsal | Lingual | Anterior | Anterior | Ventral | Dorsal Plantar | Absent |
| Set 2 | Dry | Large | Ventral | Buccal | Anterior Posterior | Posterior | Dorsal | Dorsal | Medial |
| Set 3 | Dry | Small | Dorsal | Lingual | Anterior | Anterior | Ventral | Plantar | Lateral |
| Set 4 | Dry | Small | Ventral | Buccal | Anterior Posterior | Anterior | Dorsal | Dorsal | Medial |
| Set 5 | Wet | Large | Dorsal | Buccal | Posterior | Posterior | Ventral | Plantar | Lateral |
| Set 6 | Wet | Large | Ventral | Lingual | Anterior Posterior | Anterior | Dorsal | Dorsal | Medial |
| Set 7 | Wet | Small | Dorsal | Buccal | Posterior Anterior | Posterior | Ventral | Plantar | Lateral |
| Set 8 | Wet | Small | Ventral | Lingual | Posterior | Lateral | Dorsal | Dorsal | Medial |

Each selected bone to be compressed was photographed and examined under a binocular light microscope (Leica MZ 7.5). Digested and incomplete bones were discarded to avoid fractures and weaker areas caused by digestion process. Half of selected pellets and skeletal elements were kept dry, whilst the rest were immersed in water for an arbitrary time period of fifteen days in order to examine compression under different environmental conditions. Immersion in water was considered because it could provide seasonal indications due to damp substrates. A total of four complete pellets (two dry and two wet, compressed individually) and 60 isolated skeletal elements (compressed in different sets from 5 opened owl pellets) were used in the experiment. The 60 skeletal elements were: 8 skulls, 8 mandibles, 12 femora, 8 humeri, 8 pelves, 9 astragali and 7 calcanei. The specimens selected were adult and young individuals (<1 kg, both large and small sizes) although not always a large and a small size of the same skeletal element was obtained. Two types of sediment, coarse sands (0.7 mm) and silts (0.065 mm), were chosen for testing. Each set of bones is shown in Table 1.

Pellets, both wet and dry, were pressed directly on the metal surface of the compression plates, in contrast to the isolated anatomical elements that were placed on a sediment tray.

Each set of bones was compressed in three steps as follows:

Step 1: Compression on sand

Anatomical elements selected were placed on a substratum of coarse-medium sand (0.7 mm) to show the protection this sediment type provides to small skeletal elements.

Step 2: Compression on silts

Anatomical elements were placed on silts (0.065 mm) to observe compression on a fine silty sediment typical of caves, as in Wonderwerk Cave which has served as our fossil case study (see below).

Step 3: Compression on silts without skulls

While in nature skulls are associated with other skeletal elements when trampled, the height of the skulls prevented the compression of flatter bones during the experiment. Therefore, the skulls were removed in order to ensure direct compression of the other bones and the whole SET compressed again.

Each step was carried out on four sets of dry skeletal elements and four sets of wet samples of specimens placed in different positions: Dorsal or ventral, lingual or labial, anterior or posterior, plantar or medial (see Table 1 and Figure 2).

For long bones, the resulting breakage was classified following the categories identified by Andrews [11]: Complete, proximal end, shaft and distal end. Classification of long bones was extended to facilitate the identification of breakage types using the categories described by Villa and Mahieu [17] with documentation of incipient fissures or cracks. The breakage categories proposed by Villa and Mahieu [17] were developed for long bones of large mammals, so this nomenclature has been used here without any typological meaning as it is applied to flat and small bones, such as pelves and astragali of microfauna. We have consequently named fracture angles as: Right (R), oblique (O), mixed (M); fracture outline as transverse (T), curved (C), intermediate (I) and fracture edge as smooth (S) or jagged (J). Completeness of the shaft circumference was also evaluated considering the classification

proposed by Villa and Mahieu [17]. Circumference completeness includes three categories: Bone circumference less than half of the original ①; circumference more than half of the original ② and complete circumference in at least a portion of the bone length ③ (Figure 3).

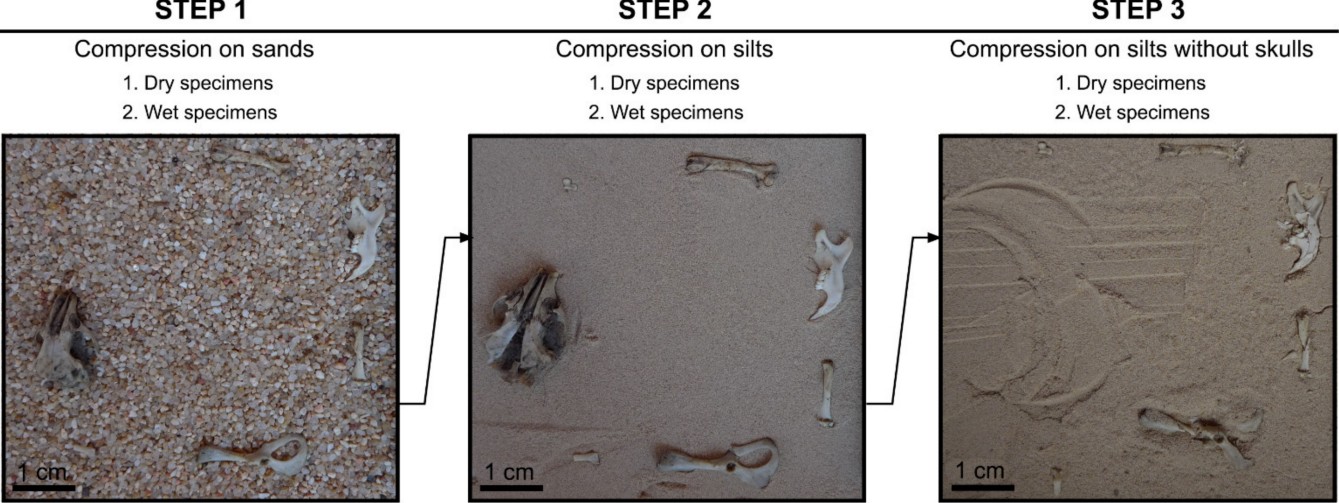

**Figure 2.** Each set of specimens as shown in this figure was compressed in (1) coarse sand substrate + all the skeletal elements, (2) silty substrate + all the skeletal elements, (3) silty substrate + postcranial elements only, repeated three times and carried out for dry and wet specimens with the skeletal elements in a different position each time.

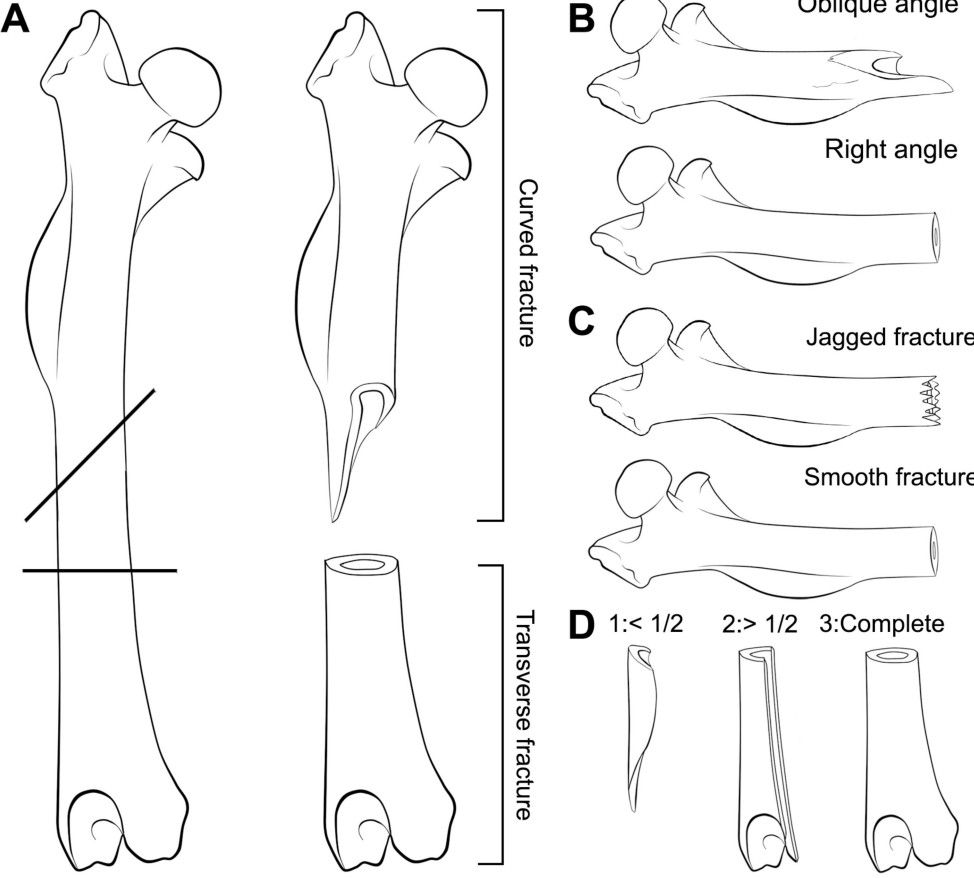

**Figure 3.** Scheme for the different fractures proposed by Villa and Mahieu [17] modified and applied to small mammals for a descriptive approach. (**A**) Fracture outline; (**B**) Fracture angle; (**C**) Fracture edge; (**D**) Completeness of the circumference.

All specimens were photographed before and at the end of each step of the experiment using an automated stereomicroscope (Leica M 205A) provided with a digital camera (Leica DFC450) and image software LAS V4.4. This procedure and the control of individual bones allowed us to discard the effects of sifting and sorting of the resulting trampled assemblage or pellet, that potentially could add to detachment of bone fragments or increase breakage.

A Fisher's Exact Test of Independence for numeric data in small samples was applied using R [32] and the library "rcompanion" [33]. This test uses a contingency table and runs an exact procedure especially for small-sized samples [34]. Fisher's Exact Test was considered to be more suitable than other tests, such chi-squared which is better for large samples. Therefore, the "simulate.*p*.value" option was used, which automatically applied a Monte Carlo simulation of the *p*-values based on 2000 replicates. The Monte Carlo approach randomly generates tables to satisfy the null hypothesis for the test and evaluate the test statistic on those tables. A significant *p*-value is considered as $p \leq 0.05$.

*Case Study of Wonderwerk Cave*

Wonderwerk Cave is a large phreatic cavity (140 m long, 11–26 m wide and 3 to 7 m high) located in the eastern flank of the Kuruman Hills, near the town of Kuruman in central South Africa, (27°50 S, 23°33 E, Figure 4). Archaeological investigations at the site started in the 1940s [35,36], but the cave was most extensively excavated by P.B. Beaumont, A.I. Thackeray and J.F. Thackeray in the 1980s [37–39]. Since 2004, it is being investigated by M. Chazan, L.K. Horwitz and their research team [40,41]. The sediment is mainly silty, sometimes sandy with some dolomite blocks that fell from the roof that are dispersed in the deposit and on the present-day cave surface.

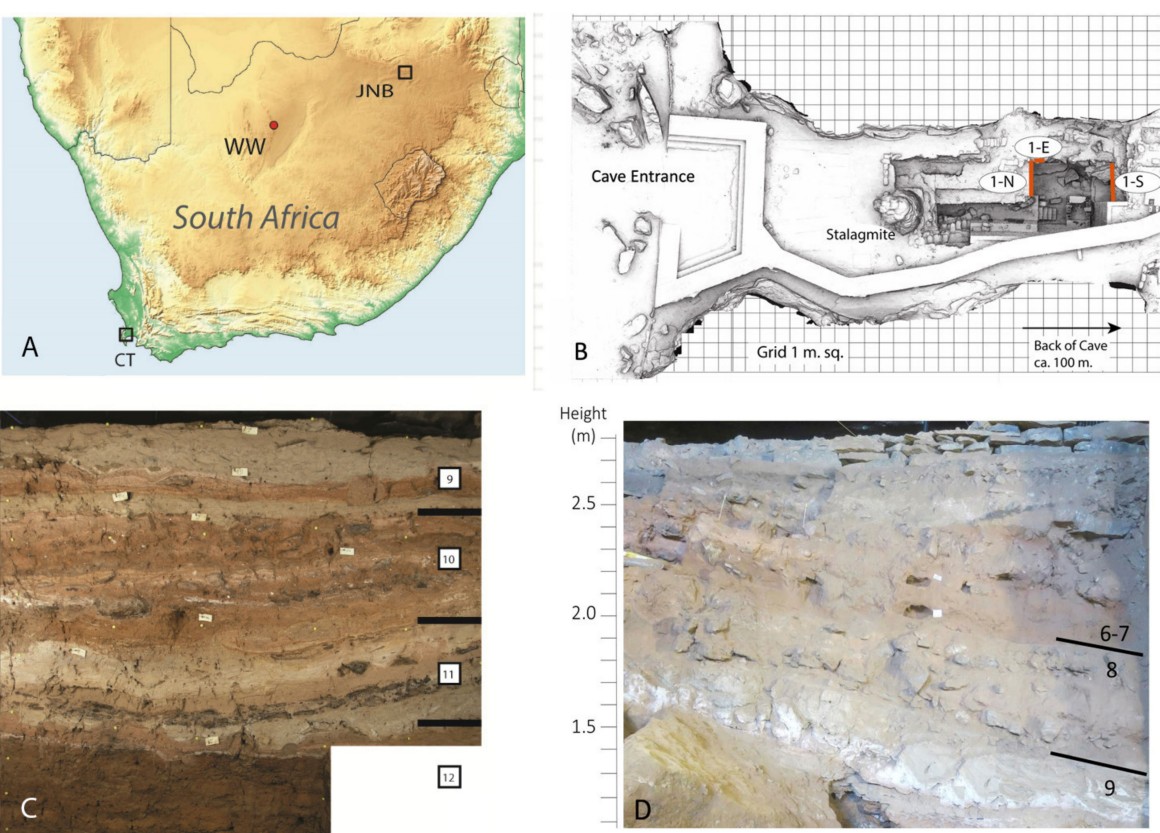

**Figure 4.** (**A**) Location of Wonderwerk Cave (WW) in South Africa (JNB = Johannesburg; CT = Cape Town). (**B**) Annotated laser scan of Excavation 1 in the cave where 1-N = North Profile, 1-S = South Profile, 1-E = East Profile (scan courtesy of the ZAMANI project, University of Cape Town). (**C**) Stratigraphy of North Profile showing Strata 12 (bottom) to 9 (top) of the sequence. (**D**) Stratigraphy of South Profile showing location of Strata 6/7.

The cave is an exceptional site yielding signs of hominin occupation spanning ca. 2 million years, from the Oldowan through to historic times [41–43]. Small mammal remains are extremely abundant in most of the layers, and in many they formed a background deposit that covered the paleo-cave floor. The physical size of the microfaunal taxa identified at this site can be considered as equivalent to the specimens used in this experiment (for taxon lists see Avery [44,45]). Taphonomic studies of these microfaunal remains have been undertaken on Oldowan and Earlier Stone Age (ESA) strata derived from Excavation 1, which is located ~30 metres from the cave entrance [30,44–47]. These publications have demonstrated the value of research on small mammal fossils at Wonderwork in identifying the predator responsible for their introduction into the cave, and for palaeoenvironmental and palaeoecological information. They have also served as sources of information on past human behaviour.

The small mammal assemblages from the oldest cave strata (St. 12 to St. 10; Oldowan through early ESA) come from the old excavations undertaken by P. B. Beaumont and were recovered by dry sieving using 1 mm mesh. Edentulous jaws and large numbers of loose teeth were the first features that caught the attention to Avery [44] who studied the taxonomy of the P. B. Beaumont collections. She concluded that the extreme rates of breakage may have been influenced by aggressive recovery and preparation procedures. Subsequently, broken edges of these fossils were evaluated by Fernández-Jalvo and Avery [46], and only 20–30% could be attributed to recent breakage. In order to contrast the incidence of recovery procedures, these old samples were compared to new samples from overlying ESA layers, Strata 6 and 7, recovered during the 2018 excavation undertaken by the Chazan and Horwitz team (Table 2). During these excavations, all sediment was collected and processed using a mechanized floatation machine. The recent sampling from Strata 6/7, whose data are reported here (Table 2), combined due to the absence of any significant differences between them, has substantially reduced breakage and loss of small mammal skeletal remains [30], but the main traits observed by Avery [44] are still present in the sample despite processing by floatation.

**Table 2.** Frequency of long bone (femora and humeri) portions and complete elements from four strata of Wonderwerk Cave site according to Marin-Monfort et al. [30].

| STRATA | St 6/7 | | St10 | | St11 | | St12 | |
|---|---|---|---|---|---|---|---|---|
| **FEMORA (Total N)** | **N = 1255** | % | **N = 3072** | % | **N = 393** | % | **N = 1451** | % |
| Complete | 353 | 28% | 216 | 7% | 43 | 11% | 72 | 5% |
| Proximal (+Prox + 1/2) | 650 | 52% | 1953 | 64% | 191 | 49% | 962 | 66% |
| Shaft | 61 | 5% | 351 | 11% | 76 | 19% | 236 | 16% |
| Distal (+Dist + 1/2) | 191 | 15% | 552 | 18% | 83 | 21% | 181 | 12% |
| **HUMERI (Total N)** | **N = 1425** | % | **N = 2975** | % | **N = 372** | % | **N = 1185** | % |
| Complete | 446 | 31% | 325 | 11% | 49 | 12% | 85 | 6% |
| Proximal (+Prox + 1/2) | 212 | 15% | 803 | 27% | 60 | 15% | 104 | 7% |
| Shaft | 100 | 7% | 505 | 17% | 85 | 22% | 258 | 18% |
| Distal (+Dist + 1/2) | 667 | 47% | 1342 | 45% | 178 | 45% | 738 | 51% |

## 3. Results

### 3.1. Compression of Pellets

Compression of pellets have more destructive effects when they are dry than when wet, with some heavily broken bones (see rectangle in Figure 5A) already visible on the pellet's surface. Wet pellets become more plastic, as well as the bones in their interior. After the experiments, the dry pellets recovered their original thickness to some degree, but the wet ones remained deformed and laterally compressed. Only two of the four pellets compressed provided the skeletal elements used here to obtain patterns of breakage (plus some other elements: Vertebrae, scapulae or tibiae). Observations of remains recovered from these two pellets are described in the Supplementary Information. The other two compressed pellets excluded from the Supplementary Information did not yield any of the

skeletal elements individually compressed to compare patterns of breakage. Bone remains still inside the wet and dry pellets are more complete than when bones were exposed directly to the force of 500 N. This is partially due to the protection provided by hair and feathers, but also because compression was exerted only once to test the experiment. This test on pellets yielded low numbers of skeletal elements and did not allow us to compare systematically compression effects.

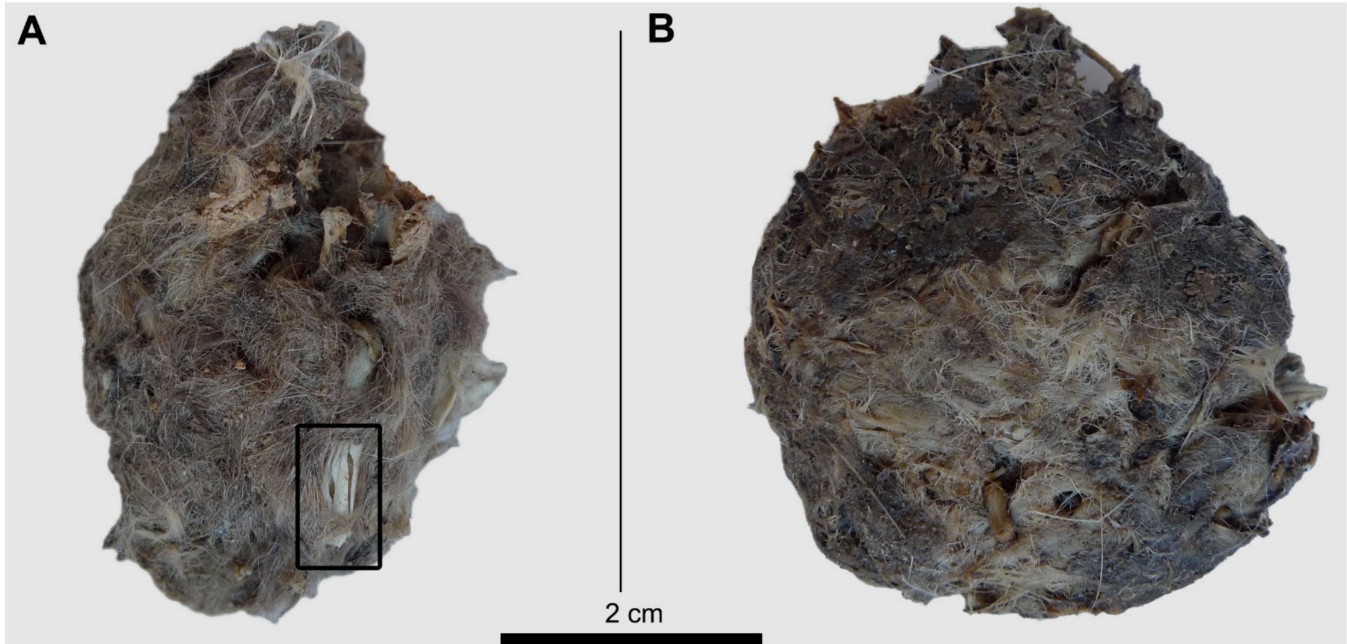

**Figure 5.** Barn owl pellets compressed under 500 N of force. (**A**) Dry pellet; (**B**) Wet pellet. The square in Figure A shows breakage as result of compression of a long bone (tibia) exposed on the pellet surface.

### 3.2. Compression of Skeletal Elements

Compression on coarse sands (Step 1) by a 500 N force did not cause significant modifications, except for skulls that opened along their sutures and for long bones that caused removal of the epiphyses, which were not fully fused to the metaphyses. For the rest of the specimens, the skeletal elements were still complete after compression.

Compression of bones on silts delivered consistent patterns, especially for cranial elements. When resting on silts, all remains showed a similar response to compression either in the first (Step 2) or in the second attempt (Step 3), except for two humeri which showed a different response in each compression attempt, even though they were not close to the skulls. These two attempts have been accounted for as two different results given more extensive breakage observed in the second compression.

Most skulls remained deformed with pieces attached together, especially in wet skulls. These pieces, however, will easily detach when the sample is processed (sifted, sorted, moved) as fossil assemblages usually are. The most destructive effects occurred when compression was on the ventral aspect and dry, while dorsal compression mainly separates the skull bones as a result of detachment of the skull sutures (Figure 6). Incisors are recovered in the interior of the alveoli (Figure 6B red arrow), and the M1 is frequently preserved in situ, with or without, the zygomatic arch attached. Frequently the M2 and M3 are detached and isolated from the maxilla (Table 3). These traits should be noted when trampling is tested in a small mammal fossil bone assemblage.

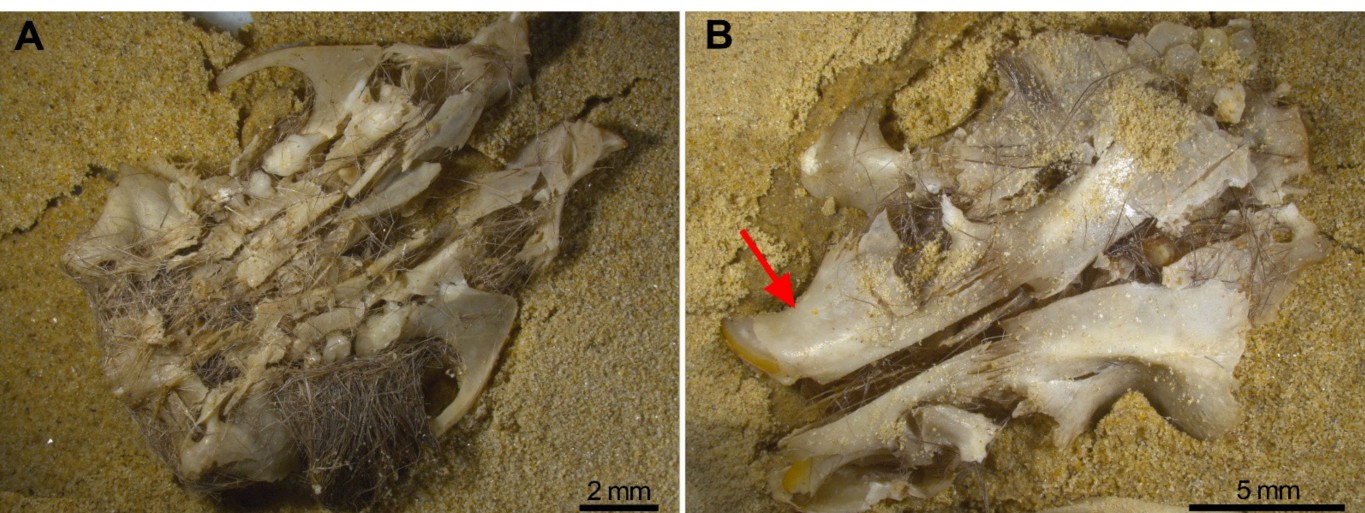

**Figure 6.** Breakage of the skull is more severe when it is compressed on the ventral aspect (**A**) rather than from the dorsal position (**B**) The red arrow points to the incisors in the interior of their sockets.

**Table 3.** Representation of cranial parts after the three steps of compression under both dry and wet conditions. Skull fragmentation appears not to be very severe, and most skulls remain attached but deformed (especially in compressed wet skulls), but any further processing of these specimens causes their disintegration into small fragments. Similarly, mandibles compressed wet remain complete. Those compressed dry may have fissures, especially affecting the ascending ramus.

| Skulls (n = 8) | N | % |
| --- | --- | --- |
| M1 + zygomatic arch | 7 | 88 |
| Incisor in socket | 8 | 100 |
| Isolated molars | 8 | 100 |
| **Mandibles (n = 8)** | | |
| Complete or longitudinal fissures | 4 | 33 |
| M1 + diastema | 7 | 58 |
| Detached molars | 4 | 33 |
| Incisor in socket | 7 | 58 |
| Detached ascending ramus | 4 | 33 |

Mandibles show the most regular breakage traits, and these specimens follow the pattern of breakage shown in Figure 7. This pattern consists of: (a) The incisors appear in the interior of their sockets (Figure 7 red arrow), (b) the M1 is attached to the diastema, (c) the ascending ramus appears almost complete but is broken and separated from the mandible, (d) M2 and M3 may be isolated and detached from the mandible or remain in situ, depending on the intensity of compression on the mandible. We, therefore, propose that when trampling aims to be tested, these four observations (Table 3) should be taken into consideration as diagnostic features of this taphonomic agent. When mandibles are wet during compression, breakage is greatly reduced and most of them remain complete. In contrast, dry mandibles show micro-fissures and small breaks compared with deformation for the wet specimens. Even wet mandibles under compression develop cracks and fissures that outline the fragments shown in Figure 7. The original position (lingual or labial) does not influence breakage.

With regard to the postcranial skeleton, pelvis survival is low independent of the position (ventral or dorsal) in which it was compressed. Compression of this element always resulted in small pieces of broken fragments (Figure 8). Amongst these fragments, the acetabulum (the articular cavity) usually survived, although when it was compressed, transversal fissures may be observed (Figure 8A, central arrow). The acetabulum may

remain attached to the bones that comprise the pelvis; more frequently it is attached to the illium, less frequently to the pubis (the branches) and never in our experiment is it attached to the ischium. Furthermore, the ischium appears always broken into several pieces (Figure 8B, Table 4).

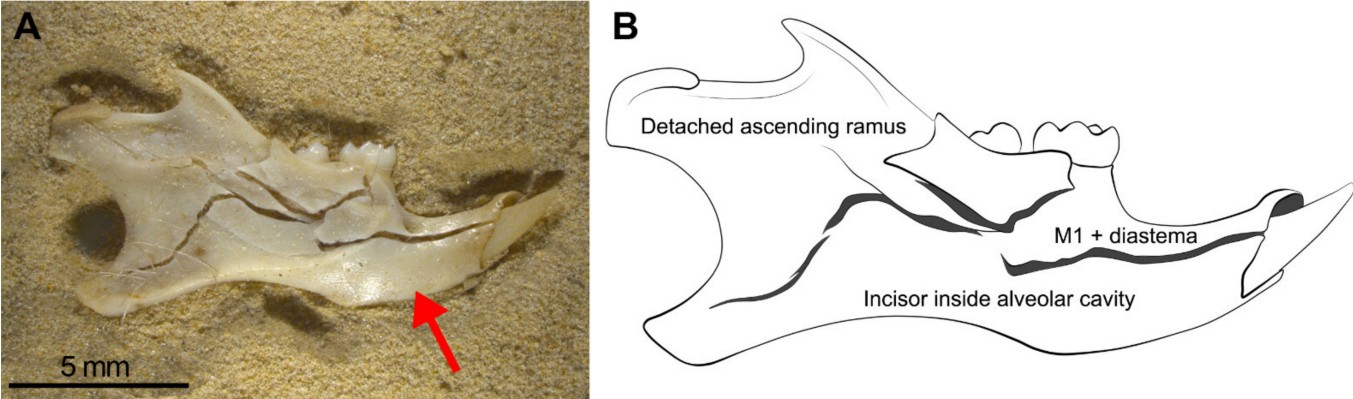

**Figure 7.** Most mandibles follow this pattern of breakage in which mandible fragments remain joined together, although with minimum movement or processing (sieving, sorting), this specimen will split into pieces. (**A**) Photo of the breakage pattern of mandibles. (**B**) Detailed scheme. The red arrow points to the incisors in the interior of their sockets.

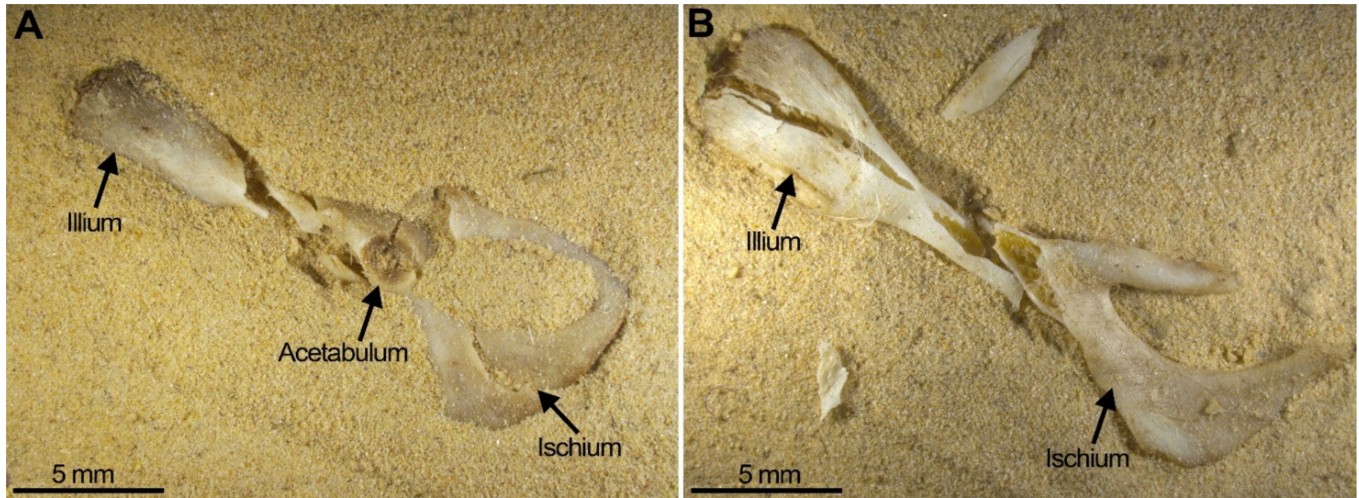

**Figure 8.** Two examples of pelvis compression: (**A**) Pelvis from SET 3 ventral view with damaged acetabulum (fissure marked by anarrow). (**B**) Pelvis from SET 2 also showing severe destruction leaving small portions of the original bone.

**Table 4.** Fragments and modifications of the pelvis when compressed, most fragments retain the acetabulum.

| Pelvis (n = 8) | N | |
| --- | --- | --- |
| Almost Complete | 1 | 13 |
| Articular Cavity (Damaged) | 2 | 25 |
| Cavity + Illium | 0 | 0 |
| Cavity + Ischium | 3 | 38 |
| Cavity + Pubis Branches | 2 | 25 |

Long bones exhibit a certain degree of variability in breakage patterns resulting from compression (Tables in the Supplementary Information). Most of the breakage types are TRJ (transverse-right-jagged) or COS (curved-oblique-smooth), while most frequently the completeness of the circumference is ③ (complete) and sometimes ① (less than half).

Femora appear more robust and resistant to breakage than humeri, with a higher number of complete elements especially when compression is on wet specimens. Despite this apparent completeness, fissures and cracks are frequent in both femora and humeri (Figure 9, red arrows). These cracks weaken attachments between the bone fragments which potentially may not last long in natural conditions or if the sample is processed by sieving and sorting. To interpret how these cracks and fissures could behave throughout their taphonomic history, we tested how they will be detached and split into two or more fragments following sample processing. Considering the additional breakage that would occur following processing, we created what we call a "compressed + processed" sample (Table 5). We then compared the sample from the compression experiment, the "compressed + processed" sample that we created and samples from the Earlier Stone Age at Wonderwerk Cave.

A Fisher's Exact Test of Independence was applied computing the *p*-value with a Monte Carlo Simulation. Results from the comparison of each level from Wonderwerk Cave with the "compressed" and "compressed + processed" sample are shown in Table 6. In general terms, St 6/7 is similar to the "compressed" sample while the other stratum, tend to be more similar to the "compressed + processed" sample (see also Figure 10). Only femora from St 12 showed differences in both the "compressed" and the "compressed + processed" sample, due to an anomalous low number of distal parts compared with the rest of the samples (Figure 10).

In our experimental data set, in contrast to all these patterns of breakage to long bones and crania, small square and compact bones, i.e., calcanei and astragali, are almost always unbroken, except in two instances out of 9 astragali (Figure 11). The broken astragali were compressed in dry conditions and broke only when placed in a dorsal position on any type of substrate (sands or silts), that possibly represents a more irregular or weaker surface. Figure 11 shows two astragali compressed in the same compression attempt, using identical force, one close to the other, one broken, the other complete. Calcanei never broke in any instance (Figure 11C).

**Table 5.** Frequency of long bones (femora and humeri) obtained following compression on current specimens. Several long bones bear cracks or fissures that may result in detachment of parts during sample processing, conforming to what we have called the "compressed + processed sample". * two out of the eight humeri showed a very different response in each compression attempt, and they have been accounted for as two different results given the strong differences observed in the third step of compression, resulting n = 10.

| | Experiment Sample | |
|---|---|---|
| **Femora (n = 12)** | Compression experiment (n) % | Compressed + processed (n) % |
| complete | (7) 41 | (3) 13 |
| proximal | (5) 29 | (9) 39 |
| shaft | (1) 6 | (4) 17 |
| distal | (4) 24 | (7) 30 |
| **Humeri (n = 10 *)** | | |
| complete | (4) 27 | (2) 10 |
| proximal | (3) 20 | (4) 20 |
| shaft | (2) 13 | (6) 30 |
| distal | (6) 40 | (8) 40 |

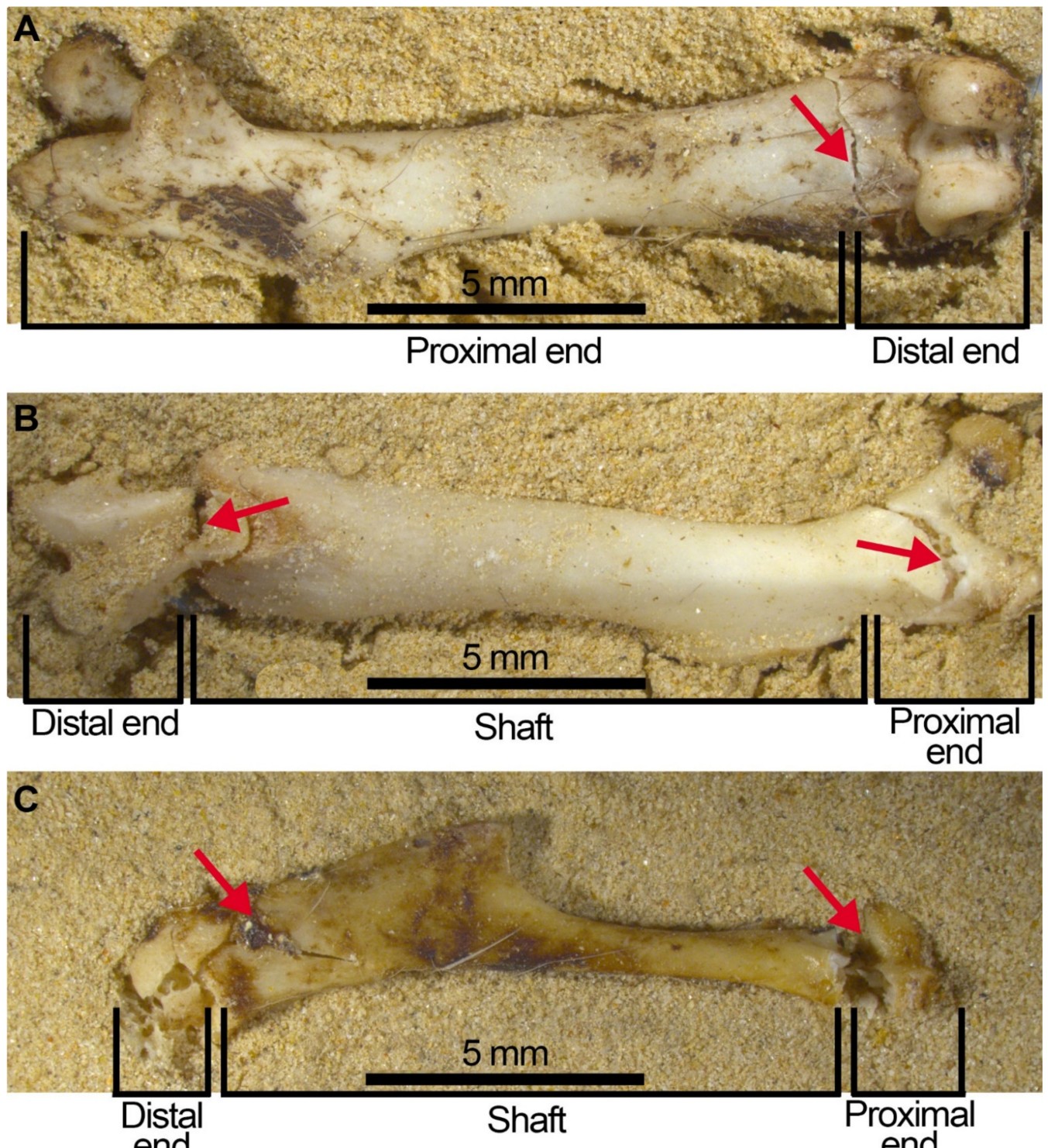

**Figure 9.** Fissures and open cracks that still join together and constitute complete bones. (**A**) complete femur circumference category ③ with a TRJ fissure on the distal end resulting in the proximal end + epiphysis segment being apart from the distal end in the compressed and processed sample. (**B**) the unfused distal end was detached from the femur's articular end during the compression experiment. The femur remains as a complete element, despite the open crack (category ③ TRJ) shown on the proximal end which will easily cause detachment of the proximal end as well as the distal end from the diaphysis. (**C**) humerus with the distal end broken leaving the proximal end and diaphysis together, although the oblique fissure on the proximal end will cause breakage of the bone into three parts: Proximal end, shaft (almost complete) and distal end.

**Table 6.** Simulate *p*-values obtained from the Fisher's Exact Test using the data showed in Table 5 compared to Wonderwerk sample assemblages published in Marin-Monfort et al. [30]. Values in bold indicate significant differences (*p*-value ≤ 0.05).

| | Femur | | Humeri | |
|---|---|---|---|---|
| **Stratum** | **Compressed** | **Compressed + Processed** | **Compressed** | **Compressed + Processed** |
| St 6/7 | 0.190 | **0.006** | 0.085 | **0.004** |
| St 10 | **0.001** | 0.052 | **0.046** | 0.518 |
| St 11 | **0.005** | 0.656 | **0.049** | 0.753 |
| St 12 | **0.001** | **0.010** | **0.002** | 0.099 |

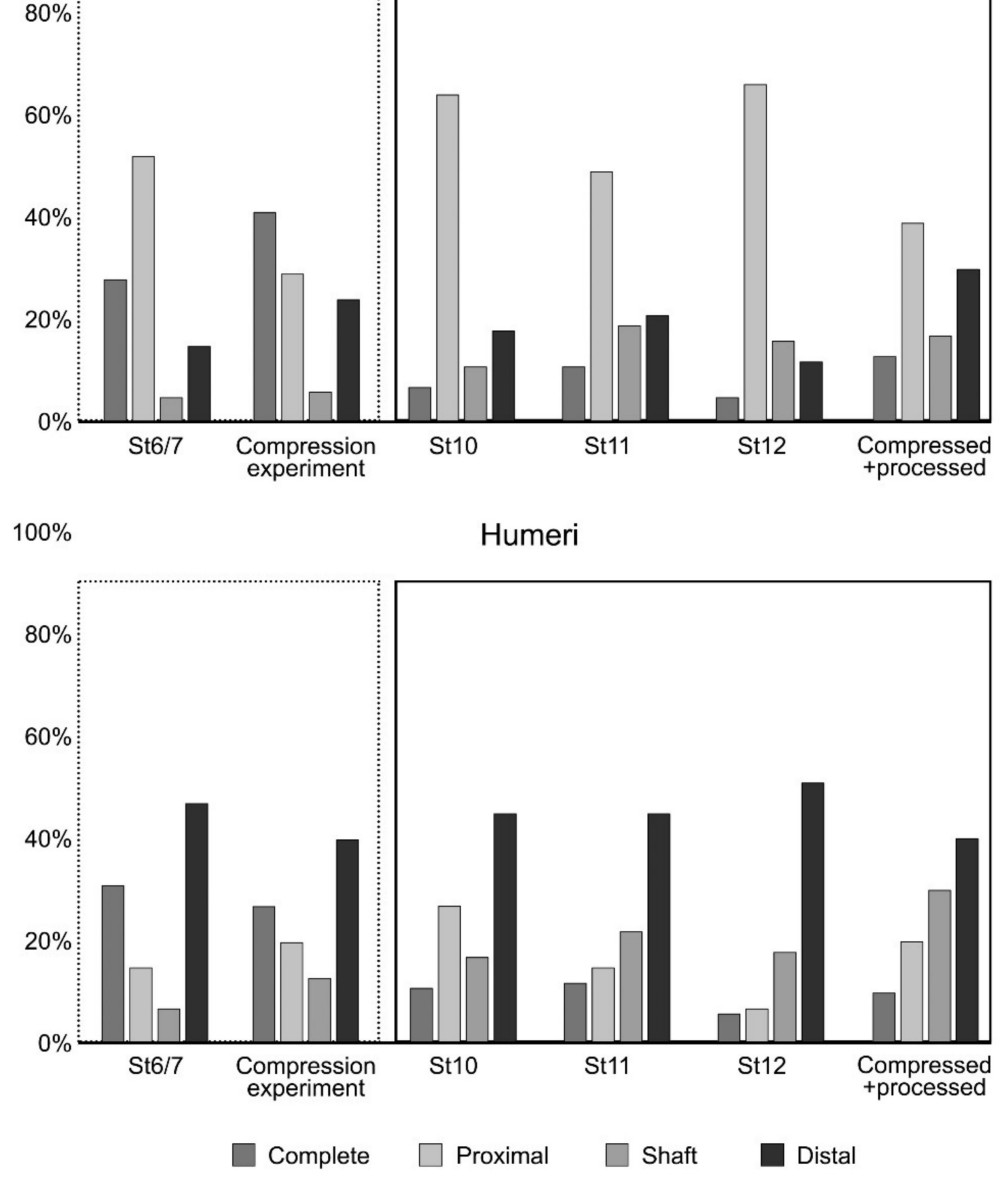

**Figure 10.** Bar diagrams of skeletal part representation in the Wonderwerk Cave strata (St. 6/7, 10, 11 and 12) compared to the fragments obtained in the compression experiment described here.

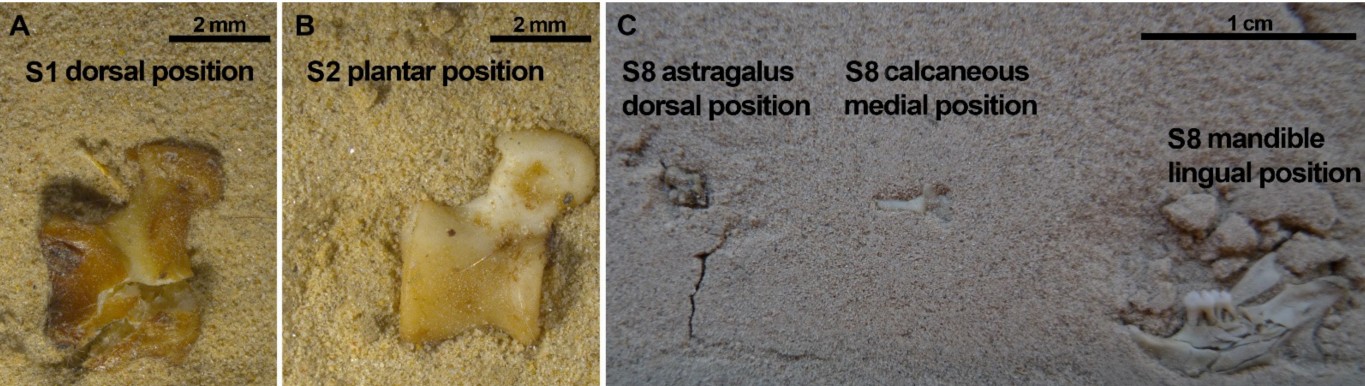

**Figure 11.** Astragali usually stayed complete after compression (75%) except for two cases in which both were positioned in a dorsal aspect and compressed dry. (**A**,**B**) show astragali from the same SET 1, resulting in one of the astragali breaking while the other remained complete during the 3rd Step of compression. Calcanei were never broken during compression (Table S5). On the right picture (**C**), SET 8, the calcaneus and the astragalus appear complete, in contrast to the mandible that is highly broken despite these elements from SET 8 were immersed in water before compression.

## 4. Discussion

Frequently, the study of taphonomy is considered to be the study of bias. Probably the reason for such an assumption is due to the inherent comparison with what should be in a site if the faunas and floras represented were complete. The loss of biological information in fossil assemblages is self-evident when observing dead bodies during processes of putrefaction. The lack of muscles and ligaments facilitates the disarticulation of the skeleton, providing important information related to the amount of time passed and weather conditions. A further source of invaluable information is the study of bone breakage and deformation. The "incompleteness" and "imperfection" of fossils, when compared to the original bones, is a source of taphonomic information that is codified in the fossils and needs to be "extracted" in order to identify the past history of assemblages and learn about site formation processes. At all sites, each fossil found can provide information on contemporaneous biotic and abiotic agents, acting either simultaneously or asynchronously on the bone remains, causing the loss of part of their original palaeobiological information and increasing the taphonomic information to be obtained [48].

The current study was initiated as result of uncharacteristically heavy breakage observed in the small mammal remains from Wonderwerk Cave. Most jaws were recurrently edentulous with a high frequency of tooth loss (the latter could easily pass through the mesh of the screens used in the original excavations), and quite extensive modern damage that was incurred due to sifting, resulting in remains that could not be taxonomically identified [44]. Despite the extensive tooth loss [44], other small-sized skeletal elements (calcanei and astragali) appeared in relative high proportions (Figure 12), suggesting that size was not the sole factor determining representation. We tested the possible involvement of highly destructive predators that could cause intense breakage to small mammals, but the predator identified at Wonderwerk Cave is *Tyto alba* (which causes minor destruction according to Andrews, [11]). Thus, this raptor could not explain the extensive destruction evidenced in these small mammal assemblages. Post-depositional abiotic agents (flowing water or falling rocks) could increase breakage in a raptor prey assemblage. However, cave roof spall does not occur in all of the cave strata that were examined here. Moreover, there is no evidence from the sedimentary record of the cave [49] that either of these factors were of any consistent intensity.

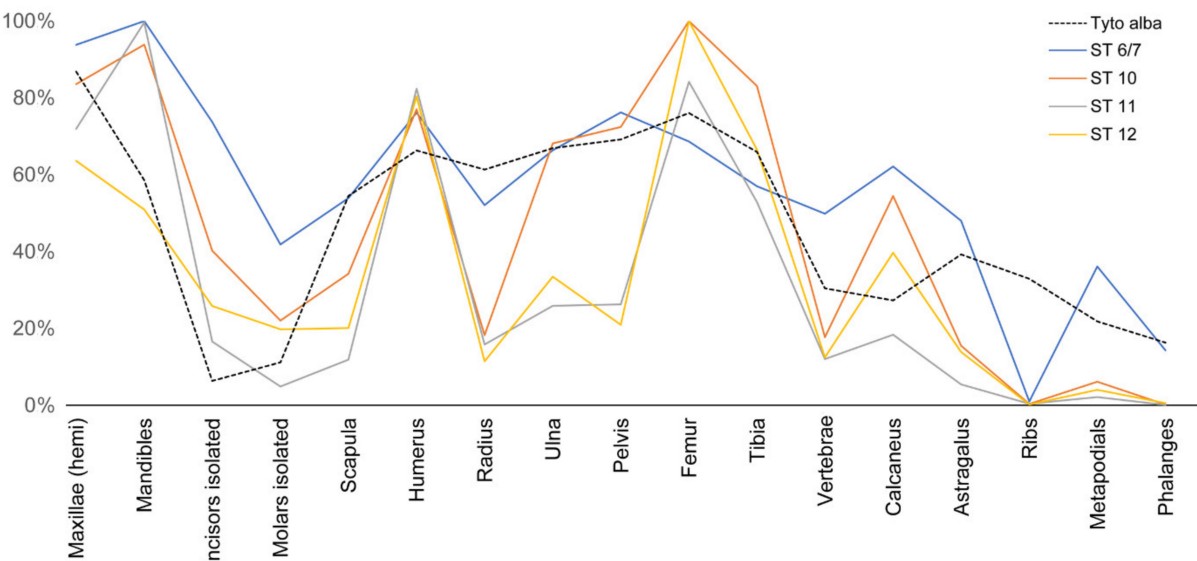

**Figure 12.** Diagram showing the relative abundance of skeletal elements in individual strata from Wonderwerk Cave (from [30]) compared to a modern barn owl (*Tyto alba*) assemblage [11]. Note that all strata show a high degree of homogeneity and have a relatively high frequency of calcanei and astragali. Note also that all samples have fewer astragali than calcanei, likely because the former incidentally broke during post-depositional taphonomic processes (as seen in our trampling/compression experiment) and possibly got lost during sample processing (i.e., sieving).

The new excavations in the Wonderwerk Cave site provided samples from St. 6/7 that exhibit greater completeness than samples from the old excavations (Table 2), but the frequency of edentulous jaws is still high. The proportion of isolated teeth (detached from jaws) and relative abundance of small-sized elements (astragali and calcanei) are even higher in the new floatation sample. In general, the traits of breakage and destruction observed in the Wonderwerk Cave samples are still higher than is expected in a barn owl assemblage.

Marin-Monfort et al. [30] observed that the destructive traits that characterised the microfaunal samples from Wonderwerk Cave were constant between layers and were associated with the persistent occupation of barn owls in the site during the ESA strata. In fact, barn owls still inhabit the cave today. Thus, the breakage patterns to the small mammal bones were incongruous with the signatures of both biotic and abiotic taphonomic agents that had been identified. We took into account the possibility of an additional factor that had not been considered before, namely that the assemblage may had been subjected to trampling, an overprint on the barn owl signature. To do that, we performed the systematic experiment described in detail in this paper examining breakage in different skeletal elements. Half of these bones were immersed in water and show a more plastic response, in contrast to dry bones that are brittle and so break more easily under compression [19,20,24].

We have confirmed the initial breakage pattern for small mammals described by Andrews [11] but added more detail to the model which can now be defined in three stages:

(A)  Breakage of skulls, reduction in numbers of maxillae. In our experiment, we have proved the pneumatic behaviour of the skull which deformed leaving a mass of bones and teeth. The bones of the skull detached into small unidentifiable bone fragments, especially when subjected to water or gravitational movements such as sifting, leaving the zygomatic arches apart from the maxillae which frequently bear the M1 in situ. Incisors were also frequently found in the interior of the alveoli, and both traits have been observed in the Wonderwerk Cave fossil assemblage. Consequently, both observations can be added as taphonomic criteria of trampling in a fossil site.

(B)   Considerable loss of teeth from the jaws leading to large numbers of isolated teeth. As seen in Wonderwerk Cave and in the experiment, both mandibles and maxillae showed detachment of molars although in mandibles the M1 remained in situ retained in the diastema portion. The incisors, as observed in maxillae, are also retained in situ (in the alveolar socket of the mandible) and in this type of assemblage it is also common to see the ascending ramus detached from the dental row and the rest of the mandible.

(C)   Considerable breakage of larger postcranial elements and some degree of loss, but no loss or breakage of smaller elements (calcanei, talli). High frequency of calcanei and astragali is one of the most representative features of microfaunal assemblages of Wonderwerk Cave site, as well as modern assemblages exposed to trampling. In addition, a high frequency of complete bones was observed when compression was undertaken on wet bones together with jagged edges, fissures and cracks, which were frequently transversal to the length of the bone.

We have obtained clear criteria with which to distinguish trampling on small mammals, by subjecting skeletal elements and pellets to compression. However, in this study our compression of pellets (Supplementary Information) may have been too mild such that the results were less destructive than found by Andrews [11]. Results of compression directly on bones, however, are highly similar. Both experiments respond to the mechanical properties of bone which behaves similar to engineering materials (e.g., ceramic, metals, rocks, building materials) and, therefore, reacts to loading and fracture following the basic principles of mechanics (e.g., [50–55]).

In general, the relevance of detecting trampling in fossil small mammal assemblages facilitates interpretation of the full history of raptor assemblages since, as at Wonderwerk, the barn owl's ingestion and digestion does not produce high bone breakage rates. Identification of trampling may then allow us to better understand how the assemblage formed. Thus, the reconstructed scenario for Wonderwerk Cave is that small mammal bones were probably trampled by the owls themselves while they nested in the cave's interior, although bones still inside the pellets could not be substantially destroyed. The owl pellets could also have been trampled by other cave visitors, including humans, thereby increasing breakage but not totally destroying the small mammal bones unless they were directly exposed to trampling (i.e., not inside a pellet). Bones from disaggregated pellets will be more exposed to post-depositional processes such as weathering and trampling than those bones in the interior of a pellet [11,56]. This scenario was recently documented in the archaeological rock shelter Álvarez 4 (late Holocene) located in the arid and cold environment of northwestern Patagonia of Argentina, where the bones from the pellets were less fractured than bone assemblages isolated in the sedimentary matrix [57].

This paper validates breakage traits obtained from experiments in a compression material testing machine against those from more actualistic experimentsof trampling by humans (e.g., [11]). There are still many experiments that need to be carried out to test other situations of trampling (e.g., pellets of other raptors and excrements of mammalian carnivores, larger and smaller prey, compression by forces around 50 N to simulate breakage by trampling in nests or latrines by predators) and so complement these patterns. There are also modern bones that were collected by one of us (PA) from monitored natural nests that potentially could have been trampled by predators. These were briefly described in Andrews [11] page 8 but as they are of interest, it is planned to study them in greater detail. In addition, it would be useful to increase the size of the experimental sample to better characterise fissures and cracks in postcranial bones and also to process these samples using sieving and floatation methods. Complementary research could extend to monitoring nesting behaviour and studying remains derived from old raptor nests.

## 5. Conclusions

-   There are distinct patterns of bone breakage and anatomical element survival that can be used to recognise the involvement of trampling in small mammal assemblages.

- Several patterns can be proposed based on the results obtained in this and previous experiments. The primary indications of trampling are: (1) Presence of edentulous jaws, (2) high postcranial breakage and (3) high frequency of complete calcanei and astragali.
- Taphonomic categories, especially with reference to cranial elements, reflect other types of breakage patterns due to compression, especially with reference to mandibles and maxillae. Long bones under compression produce characteristic fissures and cracks (transversal to the metaphysis) that may be recorded on complete or almost complete bones which indicate trampling, although these observations need further study and experiments.
- Results from the experiments we have undertaken were compared to those obtained in the actualistic experiment of human trampling (validated here) and compared to the taphonomic results of the Oldowan and Earlier Stone Age small mammal assemblages from Wonderwerk Cave (South Africa) (i.e., Strata 12, 11, 10 and 6/7). Comparison showed that trampling was an important factor responsible for the high degree of breakage observed in these assemblages. Trampling could have been caused by the predators (barn owl) themselves, as well as by other terrestrial animals or humans that visited the cave. The effects of trampling have been augmented by sifting and processing of the samples.

**Supplementary Materials:** The following supporting information can be downloaded at: https://www.mdpi.com/article/10.3390/quat5010011/s1.

**Author Contributions:** Conceptualization, Y.F.-J. and P.A.; Data curation, L.R.; Formal analysis, Y.F.-J. and L.R.; Investigation, Y.F.-J.; Methodology, Y.F.-J., F.J.F., S.G.-M., M.D.M.-M., C.I.M. and R.T.; Resources, Y.F.-J.; Supervision, P.A.; Visualization, L.K.H.; Writing—original draft, Y.F.-J., F.J.F., S.G.-M., M.D.M.-M., C.I.M., R.T., M.C. and L.K.H.; Writing—review & editing, L.K.H. All authors have read and agreed to the published version of the manuscript.

**Funding:** This project was funded by a Leakey Foundation grant to F.-J.Y. and the project CGL2016-79334-P of the Spanish National Program funded by the Ministry of Scientific Research and Innovation and the Spanish Council of Scientific Research (COOPB20287). Funding for field work at Wonderwerk Cave is provided by a grant from the Canadian Social Sciences and Humanities Research Council (SSHRC) to C.M. The Université de Rennes 1 (France) and the Museo Nacional de Ciencias Naturales (CSIC, Spain) provided the Convention de Stage n. 46360 to R.L. to do this work as practical stage of the Master M2 mention Biologie-agro-santé spécialité préhistoire, paléontologie et paléoenvironnememt in 2015. G.-M.S. has a pre-doctoral grant funded by the Universidad Complutense de Madrid (UCM) and Banco Santander (CT42/18-CT43/18).

**Data Availability Statement:** All data displayed here in tables and Supplementary Information.

**Acknowledgments:** We are especially grateful to the editors of this volume, Emmanuelle Stoetzel, Janine Ochoa and Juan Rofes for inviting us to participate in this special issue on Taphonomy, and to the reviewers whose comments have greatly improved this paper. All field work and research on the old collections of Wonderwerk Cave is undertaken under permit from the South African Heritage Resources Agency (SAHRA). Thanks also are given to IZIKO museum and curators Thalassa Mathews and Sarena Govender. We offer our warm thanks to David Morris (Head, Department of Archaeology, The McGregor Museum) for facilitating access to the Wonderwerk Cave small mammal collections in his care. Wonderwerk Cave fossil collections are currently hosted at the McGregor Museum of Kimberly (South Africa).

**Conflicts of Interest:** The authors declare no conflict of interest.

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
