# Peer review of "Understanding the Impact of Trampling on Rodent Bones"

_quaternary, doi:10.3390/quat5010011_

Round 1

Reviewer 1 Report

Understanding the impact of trampling on rodent bones” by Fernández-Jalvo and colleagues propose the results of a compression experiment on modern rodent bones from owl pellets. The paper provides news data on the subject and an application of the model to the fossil record from Wonderwerk Cave.

Several variables are tested during the experimentation (2 substratum, wet and dry, 7 different anatomical elements, adult and young, position) using a total of 60 bones. Despite the new and important data obtain by this experimentation, the authors should be more nuanced about the significance of the results - as they do in the discussion (In addition, it would be useful to increase the size of the experimental sample to better characterize fissures and cracks in postcranial bones).

Abstract : "The fragility of small mammal bones may imply breakage as a response to compression, but it has been observed that the impact of compression depends on the type of ... "Remove “but” if there is no real opposition between the two terms of the sentence.

P3 - Small mammal materials used in the experiment derive from a modern collection of pellets of barn owls kept in captivity.

  • Please precise the prey species (you explain in the result that complete pellets contain hair and feathers, so we can deduce that there are mammals and birds, but from which species?)

p 4. The specimens selected were adult and young individuals : from which taxa?

P. 5  Material and method “Most of the breakage types are TRJ (transverse-right-jagged) or COS (curved-oblique-smooth), while most frequently the completeness of the circumference is â‘¢ _(complete) and sometimes â‘  _(less than half).” .

  • This part may be remove in the result session.

P. 8 – “Compression on coarse sands (Step 1) by a 500N force did not cause significant modifications, except for skulls that were compressed on this substrate and opened along their sutures while for MANY of the long bones it caused removal of the epiphyses, which were not fully fused to the metaphyses. For the rest of the specimens, MOST of the skeletal elements were still complete after compression.

…” Mandibles show the most regular breakage traits, and MOST specimens follow the pattern of breakage shown in Figure 7.

  • Most, many is vague.  Please provide quantification or refer to a table of data.

Results: “…except for two humeri which showed a different response in each compression attempt, even though they were not close to the skulls. These two attempts have been accounted for as two different results given the strong differences observed in the second compression.”

  • Please clarify and give more explication on the subject.

P; 10 - In “Compression of pelvis always resulted in small broken fragments”.

  • Please clarify this sentence in comparison with the table 4 where 4 “Almost complete “ pelvis are mentioned. What do the percentages represent in table 4, how are they calculated? 4 of the 8 pelvis are almost complete, what mean 33% (sum of the % is more than 100).
  • Idem for Table 3 – how are calculated the % for the mandibule ? (the sum is more than 100%)

P.17 - Considerable breakage of larger postcranial elements and some degree of loss, but no loss or breakage of smaller elements (calcanei, talli). In our experimental study there was an unusually high frequency of calcanei and astragali, which is one of the most representative features of microfaunal assemblages of Wonderwerk Cave site, as well as assemblages exposed to trampling.

  • Reformulate, as example as follow : Considerable breakage of larger postcranial elements and some degree of loss, but no loss or breakage of smaller elements (calcanei, talli). High frequency of calcanei and astragali is one of the most representative features of microfaunal assemblages of Wonderwerk Cave....

- List of authors : … and Andrews, P. instead of : Andrews and P.

- Pp 2 vertebrat -> vertebrate

- p. 7 : (see rectangle in Figure 5a : no rectangle in fig 5a

- references:  in alphabetic order (Cifuentes-Alcobendas, G.; Domínguez-Rodrigo, M.

- Supplementary information, Please clarify the description from the compressed pellets. Four pellets have been compressed. What means #2 and #4? Does the list and description correspond to one ? or two pellets?

Author Response

Understanding the impact of trampling on rodent bones” by Fernández-Jalvo and colleagues propose the results of a compression experiment on modern rodent bones from owl pellets. The paper provides news data on the subject and an application of the model to the fossil record from Wonderwerk Cave.

RESPONSE TO REVIEWER'S COMMENTS ARE WRITTEN IN RED.

Several variables are tested during the experimentation (2 substratum, wet and dry, 7 different anatomical elements, adult and young, position) using a total of 60 bones. Despite the new and important data obtain by this experimentation, the authors should be more nuanced about the significance of the results - as they do in the discussion (In addition, it would be useful to increase the size of the experimental sample to better characterize fissures and cracks in postcranial bones).

Abstract : "The fragility of small mammal bones may imply breakage as a response to compression, but it has been observed that the impact of compression depends on the type of ... "Remove “but” if there is no real opposition between the two terms of the sentence. OK

P3 - Small mammal materials used in the experiment derive from a modern collection of pellets of barn owls kept in captivity.

  • Please precise the prey species (you explain in the result that complete pellets contain hair and feathers, so we can deduce that there are mammals and birds, but from which species?) Mus musculus (for small sizes) and Rattus norvergicus (for large sizes)

p 4. The specimens selected were adult and young individuals : from which taxa? Rattus norvergicus , BUT NOT ALWAYS IDENTIFIED 100%

P5  Material and method “Most of the breakage types are TRJ (transverse-right-jagged) or COS (curved-oblique-smooth), while most frequently the completeness of the circumference is â‘¢ _(complete) and sometimes â‘  _(less than half).” . deleted from Material and methods, moved to results: “ Long bones exhibit a certain degree of variability in breakage patterns resulting from compression (Tables in the Supplementary Information). Most of the breakage types are TRJ (transverse-right-jagged) or COS (curved-oblique-smooth), while most frequently the completeness of the circumference is â‘¢ (complete) and sometimes â‘  (less than half).”

  • This part may be remove in the result session. moved

P.8 – “Compression on coarse sands (Step 1) by a 500N force did not cause significant modifications, except for skulls that were compressed on this substrate and opened along their sutures while for MANY of the long bones it caused removal of the epiphyses, which were not fully fused to the metaphyses. For the rest of the specimens, MOST of the skeletal elements were still complete after compression. CORRECTED

…” Mandibles show the most regular breakage traits, and MOST specimens follow the pattern of breakage shown in Figure 7. CORRECTED

  • Most, many is vague.  Please provide quantification or refer to a table of data. CORRECTED

Results: “…except for two humeri which showed a different response in each compression attempt, even though they were not close to the skulls. These two attempts have been accounted for as two different results given the strong differences a more effective breakage observed in the second compression.” 

  • Please clarify and give more explication on the subject. (SEE ABOVE)

P; 10 - In “Compression of pelvis always resulted in small PIECES OF broken fragments”.

  • Please clarify this sentence in comparison with the table 4 where 4 “Almost complete “ pelvis are mentioned. What do the percentages represent in table 4, how are they calculated? 4 of the 8 pelvis are almost complete, what mean 33% (sum of the % is more than 100). Table and description has been simplified, in fact the table has been completely modified as it had mistaken with data copied from another skeletal element and percentages were also wrong.
  • Idem for Table 3 – how are calculated the % for the mandibule ? (the sum is more than 100%) TABLES RECORD FRAGMENTS, EACH SKELETAL ELEMENT MAY HAVE MORE THAN ONE TYPE OF FRAGMENT /EXCEPT FOR PELVES, ASTRAGALI AND CALCANEI

***P.17 - Considerable breakage of larger postcranial elements and some degree of loss, but no loss or breakage of smaller elements (calcanei, talli). In our experimental study there was an unusually high frequency of calcanei and astragali, which is one of the most representative features of microfaunal assemblages of Wonderwerk Cave site, as well as assemblages exposed to trampling.

  • Reformulate, as example as follow : Considerable breakage of larger postcranial elements and some degree of loss, but no loss or breakage of smaller elements (calcanei, talli). High frequency of calcanei and astragali is one of the most representative features of microfaunal assemblages of Wonderwerk Cave....CORRECTED

- List of authors : … and Andrews, P. instead of : Andrews and P. OK

- Pp 2 vertebrat -> vertebrateS OK

- p. 7 : (see rectangle in Figure 5a : no rectangle in fig 5a CORRECTED

- references:  in alphabetic order (Cifuentes-Alcobendas, G.; Domínguez-Rodrigo, M. CORRECTED

- Supplementary information, Please clarify the description from the compressed pellets. Four pellets have been compressed. What means #2 and #4? Does the list and description correspond to one ? or two pellets? Descriptions are based on two of the compressed pellets that contained enough skeletal material to provide reliable information

Reviewer 2 Report

see attached file

Author Response

This in an interesting article as there are few studies on the effect on trampling on small mammals bones with controlled parameters. The resulting pattern from the experimentation were applied on the fossil assemblage of Wonderwerk Cave and show an intensive breakage mainly product by animals presence in the cave.
This is a well-structured and well-written work. I recommend the publication of this manuscript, after Minor Revisions:
POINT 1. The taxonomic determination (Mus ?) of the remains used in the experiment should be added as it may have an important role in the fragmentation of the mandibles according to the size of the tooth socket.

RESPONSE 1: WE HAVE DISPLAYED THE TAXA OF THE EXPERIMENT

POINT 2. The choice of a compression force of 500N could be justified:

RESPONSE 2. EXPLAINED THE CHOICE FOR 500N IN THE PAPER

POINT 3. Maybe justified why the pellet were pressed directly on metal.

RESPONSE 3. WE SAW THE NEED TO USE SEDIMENT SUBSTRATE WHEN COMPRESSING SKELETAL ELEMENTS DIRECTLY. SEDIMENT SIZE COULD PROTECT THE SMALL SIZE OF THE SKELETAL ELEMENTS AS HAPPENED WITH SANDS. WE DID NOT SEE THIS NECESSITY IN PELLETS

POINT 4. The choice of statistical test and Monte-Carlo use should be presented in “Materials and Methodes”. Moreover, I don't know Monte-Carlo but for me a p-value≤ 0.05 gives a significant result. In your analysis it is the opposite. If this is related to the method, it should be mentioned in the “Material and Method” because the reader cannot understand why a p-value ≥ 0.05 is interpreted as significant.

RESPONSE 4. EXPLAINED BETTER THE STATISTICAL METHOD AND DESCRIBED  IN MATERIAL AND METHODS.

POINT 5. Why applicate the 3 step on each set and not multiply the sets by 3 in order to separate the influence of the substratum? it would also have allowed for a larger sample.

RESPONSE 5. I DON'T FULLY UNDERSTAND THIS OBSERVATION, BUT AT LEAST THREE COMPRESSIONS WERE MADE TO OBTAIN THE BEST PROTOCOL BEFORE  COMPRESSING THE SPECIMENS USED FOR THE DEFINITIVE EXPERIMENTS DESCRIBED HERE. 

POINT 6. Was the sediment compressed before depositing the bones? Could this have had an impact as loose sediment could absorb some of the pressure?

RESPONSE 6. WE DID SEVERAL TRIES TO CONFIRM THE EXPERIMENT COULD BE PERFORMED, SO, YES THE SEDIMENT WAS PREVIOUSLY COMPRESSED BEFORE THE REAL EXPERIMENT STARTED

POINT 7. For the figure 4 a stratigraphy of the site could be added as well as a description of the different strata. Also check the high scale of the map, it seems very low.

RESPONSE 7. STRATIGRAPHIC SECTION INCLUDED

POINT 8. In Results part there is “(see rectangle in Figure 5a)” but I can’t see this rectangle on the Figure 5a

RESPONSE 8. FIGURE 5a CHANGED ACCORDING TO SUGGESTIONS OF THE REVIEWER

POINT 9 : Table 5 the number of elements could be add to the percentage
Table 6 the caption could specify the data used, is it the data from table 2 and from the experiment?

RESPONSE 9. TABLES 5 AND 6 CORRECTED ACCORDING TO REVIEWER'S COMMENTS

POINT 10. It would be interesting to know more about the species present or at least if there are different size classes among the rodents of Wonderwerk cave.

RESPONSE 10. IDENTIFICATIONS OF STRATA FROM WONDERWERK ARE REFRRED IN THE TEXT TO PUBLICATIONS BY Avery 2007 and 2021

POINT 11. Figure 12 a multivariate analysis, such as a Principal component analysis, could provide a better interpretation of the data than this type of diagram.

RESPONSE 11: we have tried a PCA with the survival percentages, but it does not give such a good information as the diagram shown in  Figure 12, where the abundance of astragali and calcaneii is  higher than the rest of the skeletal elements which is the aim of that figure.

POINT 12. I think it has to be said that the results, although they seem clear, come from a small sample size and it is necessary to take a step back from this.

RESPONSE 12. THE SMALL SIZE OF THE SAMPLE THAT THE REVIEWER ASKS US TO DESCRIBE IS ALREADY DESCRIBED  in material and methods, showing that this is CERTAINLY small sample

POINT 13: Check the font, there is some change in the manuscript 

RESPONSE 13: FONTS HAVE BEEN CHECKED AND CORRECTED

POINT 14: I think references should be with number as for the other articles of the journal.

RESPONSE 14: REFERENCES HAVE BEEN NUMBERED AND CORRELATED AS THE JOURNAL QUATERNARY REQUESTS

POINT 15. I think there is a mistake in the end of the list of authors “Andrews and P.”

RESPONSE 15. MISTAKE OF THE AUTHOR NAME MENTIONED BY THE REVIEWER HAS BEEN CORRECTED

Reviewer 3 Report

This work if of importance and perfectly conducted. There is no doubt that this it will become a reference on the theme of “trampling”.

It also has the major interest of reconciling experimentation (compression testing in 3 steps of selected micromammal anatomical elements) and the application of concepts to a paleontological and prehistoric site (ESA layers from Wonderwerk cave, where uncharacteristically heavy breakage of micromammal bones has been observed).

From a methodological point of view, I do not see any remark to express.

Concerning the results, all cases have been taken into account: breakage of skulls, reduction in number of maxillae, great loss of teeth from the jaws, breakage of postcranial elements without loss of small elements (e. g. calcanei, talli).

Several patterns have been taken into account in the conclusion and the interpretative limits of the method are well underlined: the high degree of breakage was induced by predators themselves (even if on this point I have little doubts), as well as other larger vertebrates (mammals, including hominines) and may have been augmented by sifting and sampling.

This article can be accepted in present form.

Author Response

It also has the major interest of reconciling experimentation (compression testing in 3 steps of selected micromammal anatomical elements) and the application of concepts to a paleontological and prehistoric site (ESA layers from Wonderwerk cave, where uncharacteristically heavy breakage of micromammal bones has been observed).

From a methodological point of view, I do not see any remark to express.

Concerning the results, all cases have been taken into account: breakage of skulls, reduction in number of maxillae, great loss of teeth from the jaws, breakage of postcranial elements without loss of small elements (e. g. calcanei, talli).

Several patterns have been taken into account in the conclusion and the interpretative limits of the method are well underlined: the high degree of breakage was induced by predators themselves (even if on this point I have little doubts), as well as other larger vertebrates (mammals, including hominines) and may have been augmented by sifting and sampling.

This article can be accepted in present form.

RESPONSE: 

THANKS VERY MUCH FOR YOUR ENTHUSIATIC WORDS.  WE ALSO THINK THAT THIS TYPE OF EXPERIMENTS SHOULD BE DONE TO CONFIRM DESCRIPTIONS REFERRED TO BREAKAGE. WE WILL CONTINUE INVESTIGATING ON OTHER DIFFERENT CASES. WITH REGARD TO YOUR DOUBTS ABOUT THE CAPACITY OF OWLS TO BREAK SMALL MAMMAL BONES ACCUMULATED IN NESTS, P. ANDREWS HAS EVIDENCE OF BREAKAGE CAUSED BY RAPTORS FROM ASSEMBLAGES COLLECTED FROM BARN OWL AND EAGLE OWL NESTS (SEE Andrews, 1990, pages 8-10). OUR NEXT EXPERIMENT IS PLANNED TO APPLY A FORCE OF 10N TO 50N (EQUIVALENT TO THE WEIGHT OF PREDATORS).

Reviewer 4 Report

This study deals with the simulation of trampling on rodent bones from an experiment using the Zwick/Roell Z5.0TN machine to crush pellets (and isolated rodent bones extracted from them) of barn owls. The article describes the experiment in detail and lists the results with broken bones found in Wonderwerk Cave, South Africa. The article is very well written which facilitates fluency while reading. However, I have some observations to make regarding the experiment and the interpretation of the results:

The experiment:

The experiment described in this paper is being conducted for the first time using the Zwick/Roell Z5.0TN machine for compressed pellets of barn owls. Before using a laboratory experiment and comparing directly with bones (or fossils) found in nature, the ideal would be to first compare the results of the experiment (machine) with bones trampled on by animals or humans. Either from observations made in situ (in nature) or even from an experiment carried out in a zoo, farm, or laboratory with animals or humans (See Rozada et al. 2018 - https://doi.org/10.4000/quaternaire .8593). In other words, it would be necessary to check how closely the machine that uses uniaxial compression (we know that in nature this does not work) approaches real cases (bones that are really trampled on). If in fact the machine reproduces a considerable proportion of the marks and fractures observed in nature, a direct comparison of the results obtained with the machine and the bones found in the cave would make sense. Therefore, the direct comparison of the experiment with the cave data doesn't make much sense to me. However, this does not invalidate the authors' attempt to make this comparison, even without proving the effectiveness of the experiment with cases observed in situ.

Interpretation of Results:

I just have one observation, related to this part of the discussion:

"Thus, the reconstructed scenario for Wonderwerk Cave is that small mammal bones were probably trampled by the owls themselves while they nested in the cave's interior, although they were nested in the cave's interior, although they could not be substantially destroyed..."

I consider it unlikely that trampling by birds weighing less than 400 g, such as the Barn Owl (Tyto alba), is capable of producing any type of fracture in long bones like, humeros, femurs, pelvic girdles or even jaws. Most likely, the other alternatives pointed out by the authors, that is, humans or larger animals are really responsible for the fractures of the bones found in the cave!

Author Response

This study deals with the simulation of trampling on rodent bones from an experiment using the Zwick/Roell Z5.0TN machine to crush pellets (and isolated rodent bones extracted from them) of barn owls. The article describes the experiment in detail and lists the results with broken bones found in Wonderwerk Cave, South Africa. The article is very well written which facilitates fluency while reading. However, I have some observations to make regarding the experiment and the interpretation of the results:

The experiment:

The experiment described in this paper is being conducted for the first time using the Zwick/Roell Z5.0TN machine for compressed pellets of barn owls. Before using a laboratory experiment and comparing directly with bones (or fossils) found in nature, the ideal would be to first compare the results of the experiment (machine) with bones trampled on by animals or humans. Either from observations made in situ (in nature) or even from an experiment carried out in a zoo, farm, or laboratory with animals or humans (See Rozada et al. 2018 - https://doi.org/10.4000/quaternaire .8593). In other words, it would be necessary to check how closely the machine that uses uniaxial compression (we know that in nature this does not work) approaches real cases (bones that are really trampled on). If in fact the machine reproduces a considerable proportion of the marks and fractures observed in nature, a direct comparison of the results obtained with the machine and the bones found in the cave would make sense. Therefore, the direct comparison of the experiment with the cave data doesn't make much sense to me. However, this does not invalidate the authors' attempt to make this comparison, even without proving the effectiveness of the experiment with cases observed in situ.

RESPONSE: THE MANUSCRIPT REVIEWED HERE, ALREADY DESCRIBED THE EXPERIMENT BY P.ANDREWS DESCRIBED IN HIS BOOK 1990, OWLS, CAVES AND FOSSILS WHO DID A "REALISTIC EXPERIMENT" TRAMPLING PELLETS. THE REFERENCE MENTIONED BY THE AUTHOR IS PERFORMED ON LARGE MAMMALS IS NOT APPLYABLE TO SMALL MAMMALS THAT HAVE TO BE DONE IN A DIFFERENT WAY AS ANDREWS DID AND DESCRIBED IN OWLS, CAVES AND FOSSILS.

See Rozada et al. 2018 - https://doi.org/10.4000/quaternaire .8593).

Interpretation of Results:

I just have one observation, related to this part of the discussion:

"Thus, the reconstructed scenario for Wonderwerk Cave is that small mammal bones were probably trampled by the owls themselves while they nested in the cave's interior, although they were nested in the cave's interior, although they could not be substantially destroyed..."

I consider it unlikely that trampling by birds weighing less than 400 g, such as the Barn Owl (Tyto alba), is capable of producing any type of fracture in long bones like, humeros, femurs, pelvic girdles or even jaws. Most likely, the other alternatives pointed out by the authors, that is, humans or larger animals are really responsible for the fractures of the bones found in the cave! THERE ARE SAMPLES OBTAINED FROM NESTS (BARN AND EAGLE OWLS) COLLECTED BY PETER ANDREWS AND REFERRED IN ANDREWS (1990, PAGE 8-10) THAT ARE BROKEN IN THE CENTRAL AREA OF THE NEST, SO HE CONSIDERS THIS IS CAUSED BY THE RAPTORS. FURTHERMORE, RAPTORS USE THEIR PELLETS TO BUILD UP THEIR NESTS. WE THEREFORE START FROM THE IDEA THAT PREDATORS DO BREAK SKELETAL ELEMENTS OF MICROFAUNA TRAMPLED IN NESTS. OUR NEXT EXPERIMENT IS PLANNED TO APPLY A FORCE OF 10N TO 50N (EQUIVALENT TO THE WEIGHT OF THESE PREDATORS) AND COMPARE WITH THESE COLLECTIONS FROM NEST MONITORING.